# CRISPR-Based Genome Editing and Its Applications in Woody Plants

**DOI:** 10.3390/ijms231710175

**Published:** 2022-09-05

**Authors:** Tian Min, Delight Hwarari, Dong’ao Li, Ali Movahedi, Liming Yang

**Affiliations:** 1College of Biology and the Environment, Nanjing Forestry University, Nanjing 210037, China; 2College of Plant Protection, Northwest Agriculture and Forestry University, Yangling, Xianyang 712100, China

**Keywords:** CRISPR/Cas, gene editing, precision breeding, trait improvement, woody plants

## Abstract

CRISPR/Cas-based genome editing technology provides straightforward, proficient, and multifunctional ways for the site-directed modification of organism genomes and genes. The application of CRISPR-based technology in plants has a vast potential value in gene function research, germplasm innovation, and genetic improvement. The complexity of woody plants genome may pose significant challenges in the application and expansion of various new editing techniques, such as Cas9, 12, 13, and 14 effectors, base editing, particularly for timberland species with a long life span, huge genome, and ploidy. Therefore, many novel optimisms have been drawn to molecular breeding research based on woody plants. This review summarizes the recent development of CRISPR/Cas applications for essential traits, including wood properties, flowering, biological stress, abiotic stress, growth, and development in woody plants. We outlined the current problems and future development trends of this technology in germplasm and the improvement of products in woody plants.

## 1. Introduction

Woody plants pertain to long-term plants with highly lignified stems, many lignin cells, and moderately complex and upright stems. They are structurally defined as macrophanerophytes, bushes, and/or subshrubs appearing in different shapes. It has been proved that woody plants are assets for human health, building structure, energy, industry, landscape, food, and pharmaceuticals. Woody plants are vital in biodiversity, surface morphology, and worldwide climate [1]. Therefore, it is critical to improve the environmental resistance, production, and other traits of woody plants through genetic engineering technology to cultivate new higher-quality varieties.

Compared with seasonal herbs, woody plants have many interesting natural characteristics: Long life span, extended growth period, vast and complex genome, indefinite hereditary basis, and complex regeneration capacity [2]. Previously, woody plant breeding methods were carried out through cross-breeding techniques [3]. However, the process was slow and lengthened the production cycle. Up-to-date, modern genetic engineering breeding methods are efficient and shorten the breeding and production cycles through DNA manipulation [4,5]. Therefore, these methods are essential for modern breeding and research work in woody plants. There are remarkable advances in woody plant genetic engineering, such as biotic and abiotic stress resistance [6] and trait improvement [7]. As a core of conventional genetic engineering for plant breeding technology, transgenic technology has improved germplasm and allowed gene regulation expression analyses of target genes in plants [8,9].

Nonetheless, precise target genome editing cannot be achieved in woody plant breeding due to the problems discussed below [10]. Recently, scientists overcame the barriers of precise genome editing in poplar using optimized transformation efficiency to replace exogenous *BleoR* [11].

The recently developed gene editing technology can recognize the precise editing of specific gene loci and shorten the breeding cycle [12]. As a result, the emergence of gene editing technology provides new ideas and methods for the genetic improvement of woody plants and the research of plant gene functions. Recently, gene editing technologies have been classified into three technical systems [13]. The first is zinc finger nuclease technology (ZFN), which comprises the zinc finger protein DNA-binding domain and specific endonuclease FokI [14]. Lloyd et al. [15] transferred a vector carrying the ZFN coding gene and its target into the *Arabidopsis thaliana* genome. In T1 transgenic seedlings, ZFN gene expression was induced by heat shock, resulting in a high mutation frequency of ZFN recognition sequence, which proved that ZFN could perform specific site cleavage and induce mutation in the genome of higher plants. Subsequently, this technology was applied to target endogenous genes in woody plants, such as apples [16] and poplars [17], to produce ZFN transgenic trees. However, the application of the ZFN technique is now limited because of its off-target effects, complicated operation, and cost. Transcription activator-like effector nucleases (TALENs) are the second gene editing technology system comprising TALE protein and FokI [18]. Christian et al. [19] first applied TALENs to target the *ADH1* gene in *Arabidopsis thaliana*, resulting in the deletion of its bases. TALENs are more precise and efficient in editing than ZFNs, but the large carriers and high cost limit their applications in plants. The clustered regularly interspaced short palindromic repeats/CRISPR-associated 9 (CRISPR/Cas9) system, which was first discovered in bacteria and archaea, is the third-generation gene editing technology that shows strong gene editing ability. The CRISPR/Cas comprises CRISPR sequences and a Cas protein. In the process of genome editing with CRISPR/Cas9, the single-guide RNA (sgRNA) is formed by pairing CRISPR RNA (crRNA) and trans-activating crRNA (tracrRNA), which plays a role in targeted cutting [20]. Different domains define different Cas proteins. The Cas9 protein has six domains: Rec I, Rec II, Bridge Helix, RuvC, HNH, and the PAM-interacting domain [21]. The HNH and RuvC are nuclease domains that cleave single-stranded DNA. The HNH domain cleaves the DNA strand complementary to crRNA, while the RuvC domain cleaves the non-complementary strand [22].

Single-guide RNA (sgRNA) directs the Cas protein to cleave double-stranded DNA (dsDNA) in specific target gene sequences, creating double-stranded breaks (DSBs). Then, DNA polymerase I will be recruited to repair DSBs through non-homologous end-joining (NHEJ) and/or homology-directed DNA repair (HDR) systems [23].

Gene editing using the CRISPR/Cas system is more efficient than the ZFN and TALEN since it has increased precision, is simple to operate, cost-effective, and is multifunctional [22,24,25]. Therefore, the CRISPR/Cas technique is a milestone in genome editing technology and has various functions, including increasing the proficiency of woody plant breeding cycles, diversifying woody plant traits, and regulating insufficiencies of conventional woody plant breeding [26]. To date, the CRISPR/Csa9 technique has been used in various woody plants research. However, there is still a need for comprehension and application expansion of the CRISPR/Cas technique. In this review, we discuss the existing research on gene editing technology in woody plants, the limitations of its applications in woody plant breeding, and its future development trends.

## 2. CRISPR/Cas-Based Systems and Editors

### 2.1. CRISPR/Cas System

According to the function of Cas protein, two major classes of CRISPR/Cas system (Class 1 and 2) have been demonstrated, including their types (I–VI) and subtypes [27]. Class 1 comprises types I, III, and IV, which protect the genome of organisms from foreign invasion. In addition, it utilizes multiple Cas proteins to form a cascade complex (CRISPR-associated complex for antivirus defense) that readily cleaves the target site, causing double-strand breaks of foreign DNA. Class II comprises types II, V, and VI and functions as a single protein. Type II is widely used and was first used in human cells [28]. In the past decades, the CRISPR/Cas9 technique has been manipulated in genome editing of various plants [29,30,31], and new gene editors have been developed based on the CRISPR/Cas technique for genome editing promotion.

### 2.2. New Editors Based on CRISPR/Cas

#### 2.2.1. CRISPR/Cas12

The type V CRISPR/Cas12 system is a comprehensive defense system with up to 11 identified subtypes. Cas12a, also known as Cpf1, is a type V-A CRISPR/Cas system that is closely similar to Cas9. Xu et al. [32] first used the CRISPR/Cas12a in rice to edit the *OsPDS* and *OsBEL* genes, successfully mutating these two targets. Subsequently, Cpf1 has been widely used in plant genome editing since it does not depend on remote DNA and maintains a strategic distance from genome contamination by foreign DNA.

The Cas12b CRISPR system protein (C2c1) is a double-RNA-directed DNA endonuclease of the type V-B CRISPR/Cas system, with amino acids ranging from 1100 to 1500 and smaller than Cpf1 and Cas9 proteins [33]. However, Cas12b was unsuitable for genome editing due to its thermophilic characteristics. Cas12b requires a combination of crRNA and tracrRNA to form sgRNA targeting DNA and is more responsive to modification than Cas12a [34]. Recently, Wu et al. [35] have used two endonucleases, BvCas12b and BhCas12b v4, to generate mutant *Arabidopsis* through multiple gene editing and produced large deletions at multiple sites. However, Cas12b has a shallow off-target rate and the potential for more precise genome editing than Cas9 [36].

#### 2.2.2. CRISPR/Cas13

The type VI CRISPR/Cas system (CRISPR/Cas13) has been utilized for RNA cleavage, and Cas13a was the first RNA nuclease to be applied [37]. Abudayyeh et al. [38] found that the RNA targeting system could inhibit endogenous genes in protoplasts of *Oryza sativa* cultured in vitro. However, Cas13a has a low off-target rate compared with the traditional RNAi technology, which both have a similar RNA knockout strain to target the RNA in the nucleus. Cas13a has been applied recently in plants against viruses, such as *Turnip Mosaic Virus* (TuMV) and *Potato virus Y* (PVY) [39,40]. Moreover, the non-specific RNA restriction enzyme activity attached to the activation of Cas13a is applied to the rapid detection of nucleic acid and is used in agribusiness, character detection, problem monitoring, and distinguishing between pathogens in crop breeding [41]. The CRISPR/Cas13 may be summarized as a novel CRISPR system targeting RNA, with high targeting efficiency, involved in RNA detection, edition, knockout, tracking, and imaging [37]. The CRISPR/Cas13 system is beneficial because of its reduced operational genomic damages, regulating RNA to change the expression effect of the target gene. Therefore, it is more manageable and efficient than the traditional CRISPR gene editing techniques, such as RNAi. Cas13, as an RNA editing tool, has broad potential applications and prospects.

#### 2.2.3. CRISPR/Cas14

CRISPR/Cas14a is a compact nucleic corrosive protease that targets single-stranded-DNA (ssDNA) cleavage. It mainly binds and cleaves the target ssDNA under the direction of sgRNA. The Cas14 protein has a small amino acid length (400–700 aa) and does not require a PAM sequence. Similar to the Cas12 protein, Cas14a binds to target nucleic acid, resulting in a trans-cleavage activity. Contrary to the Cas12 protein, CRISPR/Cas14a only binds to ssDNA targets. Therefore, amplified and enriched target nucleic acids need to be treated with T7 nucleic acid exonuclease, such as amplification with phosphothiolation cation primers to ensure that only one strand of the T7 exonuclease is cleaved. CRISPR/Cas14a is an ideal tool for combating plant ssDNA viruses [42].

#### 2.2.4. Base Editing (BE)

Base editing combines the targeting function of specific DNA sequences recognized by the CRISPR/Cas system with the biochemical functions of adenine-and-cytosine deaminase to replace a single base in a target DNA sequence. Depending on the type of deaminase combined, base editing systems can be classified as cytosine base editors (CBE) and adenine base editors (ABE) [43]. The CBE (Figure 1E) system completes the single base replacement of the target site C-T (G-A): The ABE (Figure 1F) system completes a single base substitution of A-G (T-C) at the target site. Plant base editor (PBE) is a base editing system in which BE3 is applied for successful plant genome editing. Zong et al. [44] used the PBE system to edit the bases of protoplasts of crops, and the new single base editing system A3A-PBE enabled more efficient C-T single base editing in wheat, rice, and potatoes PBE [45]. The base editing system targets the desired genomic DNA (gDNA) site with no double-stranded DNA break. It uses deaminase to the target site to realize accurate single-base substitution. However, this system has disadvantages, such as PAM sequence limitation, grouping foundation reliance, and off-target impact [46].

## 3. CRISPR-Based Delivery Systems in Woody Plants

The CRISPR genome editing system requires an efficient delivery system to achieve satisfactory results [47]. One of the biggest obstacles in applying CRISPR-based gene editing in woody plants is the lack of an efficient delivery system. The current delivery methods mainly rely on agrobacterium-mediated transformation (Figure 2; Table 1). Agrobacterium-mediated transformations pose many advantages, such as low cost, developing stable transgenic plants by integrating the desired foreign DNAs, and carrying multiple binary vectors into the plant cells. However, *Agrobacterium tumefaciens* can contaminate many plant tissues, damaging the plant materials during the transformation process.

The polyethylene glycol (PEG)-mediated delivery method has been applied to the woody plant genome editing system, which can realize exogenous DNA-free cells (Figure 2; Table 1). In *Hevea brasiliensis*, the PEG was used in a delivery-mediated system of Cas protein and sgRNA pre-assembled into a dynamic CRISPR/Cas ribonucleoprotein complex (RNP) in vitro into plant protoplasts with no exogenous DNA expansions [48]. Moreover, this method has been applied to apples and grapes [49,50]. However, it has not been extensively used in woody plants because of the limitations in protoplast preparation, transformation, and regeneration techniques. Similar to the PEG method, liposomes could mediate protoplast transformation by encapsulating RNP and delivering it to the protoplasts through endocytosis, resulting in a more efficient process than PEG [51]. Therefore, establishing and improving the protoplast regeneration system of woody plants will accelerate the genome-editing-mediated breeding process. Both delivery methods require a tissue culture system, as mentioned above.

Plant genome editing can be achieved by de novo induction of meristem. Maher et al. [52] utilized the fast-processed agrobacterium coculture (Fast-TrACC) to deliver the developmental regulators WUS2, IPT, STM, and sgRNA into tobacco somatic cells to produce shoots with targeted gene modifications. It is known that growth hormones induce the formation of new meristem development, leading to the formation of buds with targeted DNA modifications, which produces transgenic sprouts in plants, including tomatoes, potatoes, and woody grapes. Research has evidenced that de novo induction of the meristem sidesteps the tissue culture process to perform plant gene editing.

First, particle bombardment is commonly used in plant gene editing systems, transmitting genetic material through rigid cell walls by a mechanical force [53] (Figure 2; Table 1). However, it is less efficient than agrobacterium-mediated methods and may also cause the rupture of foreign genes and the destruction of genomic sequences during bombardment [54,55,56]. However, this method is yet to be reported in woody plants.

Plant virus vectors have also been utilized for in vivo delivery and transient expression of foreign genes (Figure 2). Hu et al. [57] designed a barley stripe mosaic virus (BSMV)-based gRNA delivery system for CRISPR/Cas9-mediated targeted mutagenesis in wheat and corn with efficiencies of 78% and 48%, respectively. Ma et al. [58] used an engineered plant minus-strand RNA elastomeric vector to deliver efficient, stable, heritable targeted gene editing CRISPR/Cas9 endonuclease in heterologous tetraploid tobacco. This technique is advantageous since the edited plants will not contain exogenous DNA fragments because the RNA virus genome is not integrated into the host plant genome.

Nanomaterials have also been used to deliver CRISPR vectors into plants (Figure 2). Doyle et al. [59] coated GFP, Cas9, and gRNA with carbon dots (CDs) and sprayed the CDs carrying the Cas9 plasmid on wheat leaves to target the SPO11 gene. Presumably, the carbon nanotubes as new nanomaterials can deliver DNA to plant cells, such as tobacco, wheat, and cotton, and transient gene expression. This method is expected to be applied in gene editing and avoid vector fragmentation [60].

**Table 1 ijms-23-10175-t001:** Application of CRISPR/Cas system in woody plants.

	Species Name	Target Gene	Tool	Trait Performance	Transformation Method	Editing Efficiency	References
**Timber properties**	*Populus*	*4CL1*, *4CL2*	CRISPR/Cas9	Decreased lignin content, discoloration of stems	Agrobacterium-mediated	100	[61]
	*Populus tremula × P. alba*	*CSE1*, *CSE2*	CRISPR/Cas9	Reduced lignin and increased cellulose	Agrobacterium-mediated	_	[62]
	*Populus tomentosa*	*PtoMYB156*	CRISPR/Cas9	Negative regulation of secondary wall formation	Agrobacterium-mediated	48	[63]
	*Populus trichocarpa*	*PtrHSFB3-1 PtrMYB092*	CRISPR/Cas9	Reduced lignin and increased cellulose	Agrobacterium-mediated	_	[64]
	*Populus tomentosa*	*PtoDWF4*	CRISPR/Cas9	Reduced xylem development	Agrobacterium-mediated	_	[65]
	*Populus tomentosa*	*PtoDET2*	CRISPR/Cas9	Xylem development and reduced wall thickness	Agrobacterium-mediated	_	[66]
	*Populus tremula* L. × *Populus tremuloides Michx.*	*VNS*	CRISPR/Cas9	Secondary cell wall thinning	Agrobacterium-mediated	_	[67]
**Flowering**	*Hevea brasiliensis*	*HbFT, HbTFL1*	CRISPR/Cas9	bloom early	PEG-mediated	3.74–20.11	[48]
	*Apple*	*MdTFL1.1*	CRISPR/Cas9	bloom early	Agrobacterium-mediated	93	[68]
	*Pear*	*PcTFL1.1*	CRISPR/Cas9	bloom early	Agrobacterium-mediated	9	[68]
	*kiwifruit A. chinensis*	*AcCEN4*, *AcCEN*	CRISPR/Cas9	bloom early	Agrobacterium-mediated	30–75	[69]
**Biological stress**	*Populus trichocarpa*	*PtrWRKY18*, *PtrWRKY35*	CRISPR/Cas9	Melampsora resistance	Agrobacterium-mediated	_	[70]
	*Cassava*	*ncbp-1*, *ncbp-2*	CRISPR/Cas9	Cassava Brown Spot Virus resistance	Agrobacterium-mediated	91	[71]
	*Theobroma cacao*	*TcNPR3*	CRISPR/Cas9	Phytophthora resistance	Agrobacterium-mediated	27	[72]
	*grape (Vitis vinifera)*	*VvWRKY52*	CRISPR/Cas9	Botrytis cinerea resistance	Agrobacterium-mediated	5.55–27.78	[73]
	*Plasmopara viticola*	*VvPR4b*	CRISPR/Cas9	Grapevine downy mildew resistance	Agrobacterium-mediated	20.16	[74]
	*Duncan grape*	*CsLOB1*	CRISPR/Cas9	citrus canker resistance	Agrobacterium-mediated	14.29–81.25	[75]
	*Wanjincheng orange*	*CsLOB1*	CRISPR/Cas9	citrus canker resistance	Agrobacterium-mediated	11.5–64.7	[76]
	*Duncan grape*	*CsLOB1*	CRISPR/Cas9	citrus canker resistance	Agrobacterium-mediated	23.80–89.36	[77]
	*Wanjincheng orange*	*CsWRKY22*	CRISPR/Cas9	citrus canker resistance	Agrobacterium-mediated	68.2–85.7	[78]
	*Pear and apple*	*ALS*	CRISPR/Cas9 base editing	herbicide-resistant	Agrobacterium-mediated	-	[79]
**Abiotic stress**	*Populus alba var. pyramidalis*	*PalWRKY77*	CRISPR/Cas9	salt resistant	Agrobacterium-mediated	_	[80]
	*Populus trichocarpa*	*PtrADA2b-3*	CRISPR/Cas9	drought resistance	Agrobacterium-mediated	_	[81]
	*Populus*	*PdNF-YB21*	CRISPR/Cas9	drought resistance	Agrobacterium-mediated	_	[82]
**Secondary metabolism**	*Sweet orange*	*CsPDS*	CRISPR/Cas9	albinism	Agrobacterium-mediated	3.2–3.9	[83]
	*Populus*	*PtoPDS*	CRISPR/Cas9	albinism	Agrobacterium-mediated	51.7	[84]
	*Apple*	*PDS*	CRISPR/Cas9	albinism	Agrobacterium-mediated	31.8	[85]
	*Vitis vinifera* L., *cv. Neo Muscat*	*VvPDS*	CRISPR/Cas9	albinism	Agrobacterium-mediated	_	[86]
	*Cassava*	*MePDS*	CRISPR/Cas9	albinism	Agrobacterium-mediated	90–100	[87]
	*citrus*	*PDS*	CRISPR/Cas9	albinism	Agrobacterium-mediated	45.5–75	[88]
	*Coffea canephora*	*CcPDS*	CRISPR/Cas9	albinism	Agrobacterium-mediated	30.4	[89]
	*Cotton*	*GhCLA1*	CRISPR/Cas9	albinism	Agrobacterium-mediated	66.7–100	[90]
	*Green bamboo*	*PDS*	CRISPR/Cas9	albinism	PEG-mediated	12.5	[91]
	*Walnut*	*JrPDS*	CRISPR/Cas9	albinism	Agrobacterium-mediated	_	[92]
	*Populus*	*PDS*	CRISPR/Cas12	albinism	Agrobacterium-mediated	_	[93]
	*Populus*	*MYB115*	CRISPR/Cas9	Reduced proanthocyanidin accumulation	Agrobacterium-mediated	93.33–100	[94]
	*Populus tomentosa*	*PtrMYB57*	CRISPR/Cas9	Increased anthocyanins and procyanidins	Agrobacterium-mediated	_	[95]
	*Populus*	*JMJ25*	CRISPR/Cas9	Increased anthocyanin accumulation	Agrobacterium-mediated	_	[96]
	*Populus*	*UGT71L1*	CRISPR/Cas9	Partial reduction in salicylin content	Agrobacterium-mediated	40	[97]
	*Pomegranate*	*PgUGT84A23*, *PgUGT84A24*	CRISPR/Cas9	Reduced punicalagin content	Agrobacterium-mediated	_	[98]
	*Grape*	*IdnDH*	CRISPR/Cas9	Reduced tartaric acid content	Agrobacterium-mediated	100	[99]
	*tea [Camellia sinensis *(L.) *o. Kuntze]*	*CsHB1*	CRISPR/Cas9	Decrease in caffeine	Agrobacterium-mediated	_	[100]
**Growth and development**	*Jatropha curcas*	*JcCYP735A*	CRISPR/Cas9	Plant height reduction	Agrobacterium-mediated	_	[101]
	Ma bamboo (*Dendrocalamus latiflorus Munro*)	*GRG1*	CRISPR/Cas9	Plant height increase	Agrobacterium-mediated	40	[102]
	*Parasponia andersonii*	*PanHK4*, *PanEIN2*, *PanNSP1*, *PanNSP2*	CRISPR/Cas9	nodulation, bisexual flowers	Agrobacterium-mediated	48–89	[103]
	*Vitis vinifera*	*VvCCD7*, *VvCCD8*	CRISPR/Cas9	increased stem branching	Agrobacterium-mediated	66.7	[104]
	*Populus*	*BRANCHED1 BRANCHED2*	CRISPR/Cas9	Enhance the growth of shoots	Agrobacterium-mediated	_	[105]
	*Cotton*	*GhARG*	CRISPR/Cas9	Promote lateral root formation	Agrobacterium-mediated	10–98	[106]

## 4. Applications of Gene Editing Technology in Woody Plants

### 4.1. Timber Properties of Woody Plants

As a renewable biological material, wood is a necessity for human life. The formation and thickening of the secondary cell wall (lignin and cellulose deposition) are critical steps in wood formation and growth. Wood has many excellent properties and various uses. The CRISPR/Cas system has provided the functional study of genes relation and wood properties of woody plants [107].

The content of lignin accounts for about 25% of woody plants and is important in forming cell walls, providing strength and hydrophobicity to woody tissues. The CRISPR/Cas9 system was utilized to knock out the *4CL1* and *4CL2* genes from Populus with up to 100% mutation efficacy [61]. The mutant plants had a 23% reduction in lignin content, resulting in decreased lignin monophenol content. Therefore, the mutant material with a reddish-brown stem was obtained, and these results showed that the CRISPR/Cas9 system is efficient in Populus. In the monolignol synthesis pathway, CAFFEOYL SHIKIMATE ESTERASE (CSE) converts caffeoyl shikimate into caffeic acid. De Vries et al. [62] knocked out the *CSE1* and *CSE2* genes of poplar (*Populus tremula* × *P. alba*) by CRISPR/Cas9 and obtained the *cse1 cse2* double mutants, whose lignin content decreased by 35%, accompanied by growth restriction. The results showed that the *CSE* gene was an important target for improving the wood properties of woody plants. The R2R3 MYB transcription factor regulatory network can affect the formation of a secondary wall in plants. Yang et al. [63] applied CRISPR/Cas9 to knock out *PtoMYB156*, which encoded the R2R3-MYB transcription factor in *Populus tomentosa*. Their results demonstrated that lignin, xylan, and cellulose had ectopic deposition during the formation of the secondary cell wall. It was found that *PtoMYB156* negatively regulated the formation of the secondary cell wall of *Populus tomentosa*.

Tension wood (TW) is a particular xylem tissue developed and formed under mechanical or tensile stress. Its development involves the trans-regulation of secondary cell wall genes to alter wood properties and environmental adaptability. Liu et al. [64] induced TW in the stems of black cottonwood (*Populus trichocarpa*, Nisqually-1) and identified two significantly repressed transcription factor (TF) genes: Class B3 heat-shock TF (*HSFB3-1*) and *MYB092*. They knocked out *PtrHSFB3-1* and *PtrMYB092* genes by CRISPR/Cas9, which condensed the mutant lignin and high cellulose content. These results verified that these TFs played a regulatory role in the biosynthesis of cell wall components during TW formation. 

Brassinosteroids (BRs), as plant steroid hormones, play a regulatory role in xylem development. *DWF4* gene is a related gene in brassinosteroid biosynthesis. Shen et al. [65] knocked out the *PtoDWF4* gene of *Populus tomentosa* using CRISPR/Cas9, which reduced the lignin content by 8.79–11.67%. Their results demonstrated that *PtowDWF4* positively regulates xylem development during wood formation. The study analyzed the critical gene for xylem development in poplar, laying a solid foundation for cultivating poplar varieties with high-yield wood in the future. Moreover, *DET2* is an important rate-limiting enzyme in the BR biosynthesis pathway. Fan et al. [66] improved the bioethanol yield of lignocellulose from *Populus tomentosa* by regulating brassinosteroid biosynthesis. The secondary cell wall of the *PtoDET2* mutant produced by CRISPR/Cas9 became thinner, and the ratio of cellulose crystallinity index (CRI), degree of polymerization (DP), and hemicellulose increased. These findings showed that *PtoDET2* had a positive regulatory effect on xylem development, cell wall biosynthesis, and modification of poplar. The *VNS* gene family plays a critical regulatory role in secondary wall synthesis. Takata et al. [67] knocked out four *VNS* genes of *Populus tremula* L. × *Populus tremuloides Michx*. using the CRISPR/Cas9, and secondary cell walls were not formed in xylem wood fibers, xylem ray parenchyma cells, and phloem fibers in the four mutants of *vns09/vns10/vns11/vns12*, but only some wood fibers near the vessel element showed secondary cell wall deposition, indicating that *VNS* genes in *Populus* coordinated the regulation of the formation of secondary cell walls.

### 4.2. Flower Development of Woody Plants

Flowering is necessary to regulate the reproductive and genetic abilities of higher plants. This trait is delimited by genes and a complex molecular genetic network. To reach the flowering stage, woody plants undergo a long juvenile period. The CRISPR-based gene editing technology can be used to study the genes related to woody flowering, flowering traits, nutrition regulations, and the reproductive growth of woody plants to improve yield or shorten the breeding cycle.

The FLOWERING LOCUS T (FT) protein is the flowering factor signal for flowering initiation, while the TERMINAL FLOWER1 (TFL1) protein acts as an anti-flowering pigment to inhibit flowering. Fan et al. [48] knocked out the *HbFT* and *HbTFL1* genes in *Hevea brasiliensis* by CRISPR/Cas9 and obtained plants with delayed and early flowers. Charrier et al. [68] knocked out the *TFL1* gene of pear and apple by CRISPR/Cas9. Their results disclosed that 93% and 9% of the apple and pear mutant lines, respectively, show early flowering. Kiwifruit is dioecious during a long juvenile period. *CENTRORADIALIS* (*CEN*)-like genes have been identified as potential flowering repressors. Varkonyi-Gasic et al. [69] knocked out the *AcCEN4* and *AcCEN* genes of kiwifruit *A. chinensis* by CRISPR/Cas9, leading to early mutant blooming and the production of more compact fruit. The above research results based on the CRISPR/Cas9 gene editing system enable the creation of new forest germplasm with diverse flower development cycles and the shortening of breeding cycles of perennial woody plants.

### 4.3. Improvement of Stress Resistance

Biotic and abiotic factors reduce the development and growth of woody plants. The CRISPR-based gene editing system enabled the breeding of resistant varieties of woody plants.

#### 4.3.1. Biotic Stress

*Melampsora* can cause poplar rust disease, one of the most widely distributed and most harmful diseases of poplar. Salicylic acid-mediated pathways are essential for plant defense against pathogens. WRKY is one of the salicylic acid response transcription factors. Jiang et al. [70] used the CRISPR/Cas9 system to knock out the genes of *PtrWRKY18* and *PtrWRKY35* in *P. trichocarpa* and then inoculated *Melampsora*. Severe symptoms appeared on the leaves of the mutant plants. The results support the SA-mediated signal participation in poplar *Melampsora* resistance. Cassava brown streak virus is an RNA virus that causes the cassava brown streak disease, and it restricts the growth of plants. Diseases caused by this virus require the interaction between viral genome-linked protein (VPg) and host eukaryotic translation initiation factor 4E (eIF4E). Gomez et al. [71] used CRISPR/Cas9 to edit cassava eIF4E isoforms *ncbp-1* and *ncbp-2* genes and obtained *ncbp-1* and *ncbp-2* single and double mutants. After inoculation with the cassava brown streak virus, it was found that the incidence of cassava brown streak disease was significantly reduced. The disease symptoms were weakened in the double mutant plants. *Phytophthora* is a type of plant pathogenic bacteria, mostly a facultative parasite. Fister et al. [72] used CRISPR/Cas9 to knock out the *TcNPR3* gene of the *Theobroma cacao*, and the resistance of the mutant to the *Phytophthora tropicalis* infection was enhanced.

*Botrytis cinerea* is a fungal disease that can attack flowers, fruits, leaves, and stems. *VvWRKY52* gene has been proved to play a role in biological stress response. Wang et al. [73] used CRISPR/Cas9 technology to knock out the *VvWRKY52* gene of grape (*Vitis vinifera*). The inoculation experiment of *Botrytis cinerea* indicated that the mutant strain remarkably improved the resistance to *Botrytis cinerea*, and the double allele mutant strain had increased resistance than the single allele mutant strain. Grapevine downy mildew is a significant oomycete disease caused by *Plasmopara viticola* and is harmful to grape production. Pathogenesis-related (PR) proteins comprise a large class of plant disease defense proteins. Li et al. [74] used CRISPR/Cas9 to knock out *VvPR4b* gene in grapes. The mutant strain was more sensitive to *Plasmopara viticola*, which proved that *VvPR4b* has the disease resistance to *Plasmopara viticola* in grapes.

*Xanthomonas citri* subspecies citri (Xcc) is the main pathogenic factor of citrus canker, and *CsLOB1* is the susceptible gene of citrus canker. Jia et al. [75] used CRISPR/Cas9 to edit the *PthA4* effect binding element in the type I *CsLOB1* gene promoter in Duncan grapefruit, and the mutant showed slight relief from citrus canker symptoms. Peng et al. [76] edited the *CsLOB1* gene promoter in Wanjincheng orange (*Citrus sinensis* Osbeck) using the CRISPR/Cas9 technique. Moreover, mutant lines showed high resistance to citrus canker. The results prove that the CRISPR/Cas9-mediated promoter editing of *CsLOB1* is an efficient strategy for generating canker-resistant citrus cultivars. Jia et al. [77] used the newly developed CRISPR/Cas12a editor in the *CsLOB1* gene of Duncan grapefruit. Here, mutant plants with resistance to citrus canker were obtained, and the resulting biallelic mutant probability was lower than CRISPR/Cas9 editing. In addition to editing the *CsLOB1* gene, Wang et al. [78] used CRISPR/Cas9 to knock out the *CsWRKY22* gene in Wanjincheng orange (*Citrus sinensis* (L.) Osbeck), reducing the susceptibility to canker disease. All the above results are helpful in cultivating new woody varieties with disease resistance.

Herbicides are potentially harmful to woody plants when used to control the weed. Therefore, the prime needs to develop herbicide-resistant assortments. Acetolactate synthase (ALS) confers resistance to the herbicide chlorsulfuron. Malabarba et al. [79] used base editing to create an amino-acid substitution on *ALS* genes in apple and pear, which resulted in the non-binding of the herbicide chlorsulfuron. Thus, the mutant obtained resistance to the herbicide chlorsulfuron.

#### 4.3.2. Abiotic Stress

Drought, high salt, and other environmental factors affect the growth and yield of woody plants. *WRKY* gene plays a vital role in regulating the response of woody plants to salt stress. Jiang et al. [80] used CRISPR/Cas9 to mutate the *PalWRKY77* gene of *Populus alba var. pyramidalis,* which increased the salt tolerance of poplar plants. Suggesting that *Populus* can recruit the abscisic acid (ABA) signaling pathway to adapt to a saline environment. Additional research revealed that *PalWRKY77* negatively regulates the ABA-responsive gene and relieves ABA-mediated growth inhibition, which causes *Populus* to be more sensitive to salt stress. AREB1-ADA2b-GCN5 ternary protein complex can be used for drought response and tolerance of *Populus* species by coordinating histone acetylation and gene activation, which is mediated by transcription factors. Li et al. [81] used CRISPR/Cas9 to knock out the *PtrADA2b-3* gene of *Populus trichocarpa*, significantly reducing drought tolerance in mutant plants. Nuclear factor Y (NF-Y) transcription factor is important in regulating plant growth, development, and stress response. Zhou et al. [82] used CRISPR/Cas9 to knock out the *PdNF-YB21* gene of *Populus*, decreasing the root growth of the mutant plants and drought tolerance. In contrast, overexpression of *PdNF-YB21* in *Populus* promoted root growth and improved drought resistance, evidencing that the NF-Y transcription factor plays an essential role in the drought resistance of poplar.

### 4.4. Secondary Metabolism

Plants create abundant secondary metabolites that regulate growth and development, increase resistance, enhance regeneration components, and increase nutritional ability.

Carotenoids are involved in the light-harvesting stage of photosynthesis and antioxidant functions in woody plants and provide excellent color to stems, leaves, flowers, fruits, and other organs [108]. Phytoene dehydrogenase (PDS) is the first rate-limiting enzyme in the carotenoid synthesis pathway. It has been reported that a mutant with an albino phenotype was obtained by knocking out the *PDS* gene using the CRISPR/Cas9 and Cas12 systems, demonstrating the vital role of the *PDS* gene in the carotenoid synthesis pathway. The CRISPR/Cas9 technology was applied to knock out the *PDS* gene to produce albino plants in sweet orange [83], poplar [84], apple [85], grape [86], cassava [87], citrus [88], coffee [89], cotton [90], green bamboo [91], walnut [92], and other species. Moreover, An et al. [93] used the CRISPR/Cas12 to knock out the *PDS* gene of poplar and obtained albino phenotypic plants.

Anthocyanin, a water-soluble natural pigment in plants, participates in the development and defense of plants. Proanthocyanins are polyphenols, and anthocyanins are flavonoids. During heat, the procyanidins produce anthocyanins in an acidic medium. Wang et al. [94] knocked out the poplar *MYB115* gene by CRISPR/Cas9, and the accumulation of procyanidins in mutants decreased. The results indicated that the *MYB115* gene was involved in the positive regulation of procyanidins biosynthesis. Wan et al. [95] knocked out the *PtrMYB57* gene of *Populus tomentosa* with CRISPR/Cas9 system, resulting in increased anthocyanin and proanthocyanidin in the plant. *JMJ25* is a histone H3K9 demethylase gene. Fan et al. [96] knocked out the *JMJ25* gene of *Populus* by CRISPR/Cas9 system, which encodes histone H3K9 demethylase. The mutant strain produced ectopic anthocyanin accumulation and increased the anthocyanin biosynthesis gene expression.

The salicylate is broadly covered within the bark and removes different varieties of salix in poplar plants, which may play a crucial part in resisting the herbivores. UDP-glucose-dependent glycosyltransferase (UGT) is a widespread enzyme in plants that plays several roles in secondary plant metabolism, hormonal, and other small molecule detoxification. Fellenberg et al. [97] knocked out the *UGT71L1* gene related to the synthesis of salicylate from poplar by CRISPR/Cas9, resulting in a partial reduction in salicylate content. Pomegranate is rich in phenolic metabolites. Chang et al. [98] used CRISPR/Cas9 to knock out the two UDP-glycosyltransferase genes *PgUGT84A23* and *PgUGT84A24* in pomegranate (*Punica granatum* L.), thus reducing the content of phenolic substance angoranin in pomegranate. Ren et al. [99] knocked out the tartaric acid (TA) synthesis key gene *IdnDH* in grapes using CRISPR/Cas9, which remarkably reduced the TA content in the mutant and confirmed the feasibility of applying the technology to grape genome transformation.

Caffeine is a plant alkaloid, and its anabolism in tea is regulated by the N-methyltransferase gene (NMT), an essential enzyme gene. Ma et al. [100] cloned *CsHB1* from tea plant (*Camellia sinensis* (L.) o. Kuntze) by a single yeast hybrid using *yhNMT1* as bait. Using CRISPR/Cas9 technology, the *CsHB1* gene was effectively mutated in transgenic tea callus. This reduced the expression of *CsHB1* by 65%, downregulating the expression of N-methyltransferase gene *yhNMT1*, which catalyzes caffeine synthesis in tea. Consequently, this reduced the caffeine content in callus and its caffeine accumulation by 97.5%. This study enhanced the understanding of the regulatory mechanism of caffeine biosynthesis in tea.

### 4.5. Growth and Development

Plant hormones can affect physiological processes, such as germination, rooting, and fruiting, and play an important role in regulating the growth and development of plants. Common hormones include auxin, gibberellin, cytokinin, abscisic acid, ethylene, brassinosteroids, and strigolactone. Cai et al. [101] knocked out the cytokine synthesis-related gene *JcCYP735A* of *Jatropha curcas* by CRISPR/Cas9, decreasing the cytokinin content in leaves. The plants indicated apparent growth retardation, and the height was only a quarter of the wild plants. Ye et al. [102] knocked out a gibberellin response gene *GRG1* in hexaploid Ma bamboo (*Dendrocalamus latiflorus Munro*) by CRISPR/Cas9, resulting in a significant increase in plant height. It is of great significance to study the rapid elongation and growth of Gramineae bamboo.

Tropical tree *Parasponia andersonii* has the ability of nodulation and nitrogen fixation. Zeijl et al. [103] knocked out *PanHK4*, *PanEIN2*, *PanNSP1*, and *PanNSP2* genes of *Parasponia andersonii* using the CRISPR/Cas9, which control cytokinin, ethylene, or strigolactone hormone networks, respectively. They found that the *PanNSP1* and *PanNSP2* genes are crucial for nodule formation, while *PanHK4* mutation leads to a decrease in the cambium activity of stem, and *PanEIN2* mutation leads to gender differentiation.

Strigolactone (SLs) is a newly discovered plant hormone that can inhibit branching in woody plants. *CCD7* and *CCD8* are SL biosynthesis genes. Ren et al. [104] edited *VvCCD7* and *VvCCD8* genes in *Vitis vinifera* using CRISPR/Cas9. The *ccd8* mutant had more branches than the corresponding wild-type plant, and the results emphasized the critical role of *VvCCD8* in controlling grape stem branches. The TCP-type transcription factors *BRANCHED1* and *BRANCHED2* shape the plant architecture by suppressing bud outgrowth. Muhr et al. [105] used the CRISPR/Cas9 system to knock out the transcription factors *PcBRC1-1* and *PcBRC2-1* of *Populus*. The branches in the mutant plant significantly increased, and the *pcbrc2-1* demonstrated a higher bud growth rate. Arginine in plants has the function of storing nitrogen nutrition. Wang et al. [106] knocked out two *GhARG* homologous genes in upland cotton by CRISPR/Cas9, which inhibited the arginase activity. Nonetheless, they activated nitric oxide synthase activity, inducing arginine to synthesize more nitric oxide (NO) and promoting the formation of lateral roots, thereby potentially promoting the growth of the whole cotton plant and finally increasing the fiber yield. To summarize the manipulations of the CRISPR/Cas system in woody plants discussed above, Figure 3 below sheds light on the key features of the CRISPR/Cas gene editing.

## 5. Conclusions

Forest trees have significant financial and environmental benefits within human society; therefore, it is crucial to carry out germplasm development and gene editing on woody plants. The CRISPR/Cas system has been manipulated to investigate crucial characteristics of woody plants, such as wood properties, blooming, natural product quality, and resistance. These are prime focal points in forestry breeding, trait development and alteration, and quality enhancement. With the improvement and innovation of CRISPR/Cas gene editing technology, some other proficient and accurate gene editing systems are constantly emerging. The above-discussed researches illustrate the application of the CRISPR/Cas9 technology, using current gene editing techniques in woody plants. Certain newly developed editors, such as single base editing and CRISPR/Cas12, are also applied, and further novel editors will be applied in woody plants in the future, such as CRISPR/Cas13, CRISPR/Cas14, and prime editor. However, there is still a need to improve the application of these tools in woody plants. Broadening the application of these modern advances in woody plants will encourage the acceleration of the qualitative research process and advancement of assorted woody plants.

CRISPR/Cas system knocks out plant genome and creates gene additions and deletions at the specific site. The effectiveness of target inclusion and substitution primarily involves constraining its application in plant genome research and breeding.

Establishing a more efficient technical system for inserting and replacing plant genome fragments is urgent. CRISPR/Cas9 technology has a broad application prospect, but there are still particular shortcomings, such as PAM sequence dependence and distance and editing efficiency. The off-target phenomenon can be reduced using the appropriate Cas protein or improving the specificity of the sgRNA sequence.

In woody plants, gene editing innovation inquiries primarily employ agrobacterium-mediated or PEG-mediated change innovation as a delivery system. Due to the confinement of the natural characteristics of different woody plants, these delivery systems are limited to a few plant species or tissues. The strategy of inducing meristem from scratch based on Fast-TrACC has comprehended the editing of plant genes in grapes, avoiding tissue culture. It is expected to develop further into a new strategy to overcome the bottleneck of editing plant genes. At the same time, the established gene gun method and virus vector delivery method have exposed their high efficiency and convenience in the gene editing system of model plants and herbs. Nonetheless, a mature technical system must be established in woody plants, especially forest trees. The nanomaterial-based delivery system can combine CRISPR/Cas systems to create genetically engineered plants without transgene integration, while selecting and realizing efficient delivery according to the characteristics of nanomaterials. The above-discussed delivery strategies are predicted to be the future development trend of plant gene editing, but whether they are equally applied to woody plants remains to be confirmed by research.

Compared with herbaceous plants, CRISPR/Cas system is still in the primary stage of establishing a gene editing system in most woody plants. There are still many difficulties and challenges in applying CRISPR/Cas system to woody plants. The main reasons are as follows: First, woody plant genomes are huge, highly heterozygous, and repetitive. Second, there are few woody plants with completed genome sequencing at present, and the genomic data of forest species recorded in the target site design and analysis platform is not perfect, which leads to problems, such as the inability to design gRNA. Finally, most woody plants have not yet established a stable, efficient, and time-consuming genetic transformation system. Developing protoplasts of woody plants to plan tissue culture recovery systems and progressing the effectiveness of single-cell change will help accelerate the era of high-quality strains by gene editing. It is expected that with the research of genes related to imperative characteristics and infections of woody plants, gene editing innovation based on CRISPR/Cas will be an essential device for genetic engineering research in woody plants.

## Figures and Tables

**Figure 1 ijms-23-10175-f001:**
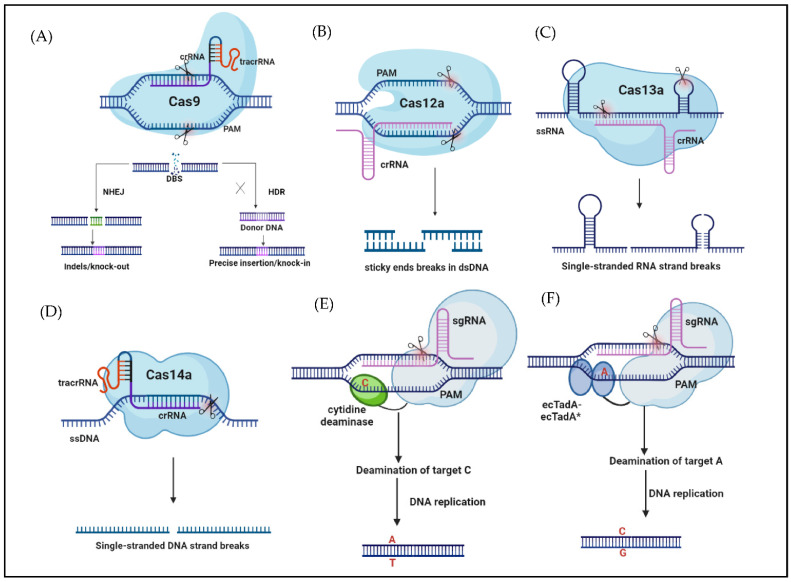
Representative schematic diagrams of CRISPR/Cas editing. (**A**) Cas9 applies the PAM and sgRNA to cleave the target DNA and produce a DBS, which is repaired by NHEJ or HDR; (**B**) Cas12a uses a RuvC domain under the guidance of crRNA without the participation of tracrRNA to cleave dsDNA, producing a sticky end; (**C**) Cas13a targets RNA in the nucleus; (**D**) CRISPR/Cas14a targets ssDNA cleavage under the direction of sgRNA, and does not require a PAM sequence, producing SSB; (**E**) CBE system compliments the single base replacement of the target site C-T (G-A), cleaves a single target locus, and produces a staggering cut; (**F**) ABE system compliments a single base substitution of A-G (T-C), cleaving at the targeted loci, and displaces the DNA fragment, leaving a staggering end in dsDNA (Provided by BioRender.com; accessed on 14 March 2022).

**Figure 2 ijms-23-10175-f002:**
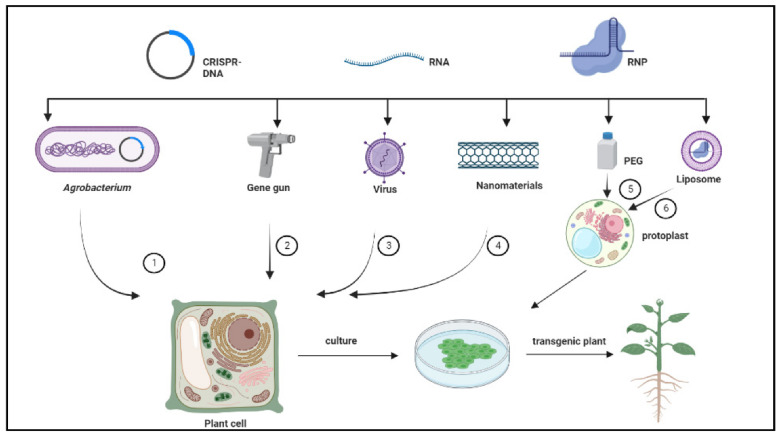
CRISPR/Cas system delivery technology, transferring foreign recombinant DNA by the CRISPR/Cas9. From the left side, (1) agrobacterium tumefaciens, (2) gene gun, (3) virus vector, (4) nanoparticles, (5) PEG, and (6) liposome carrying the targeted gene into the plant cell, which is further cultured (Provided by BioRender.com; accessed on 14 March 2022).

**Figure 3 ijms-23-10175-f003:**
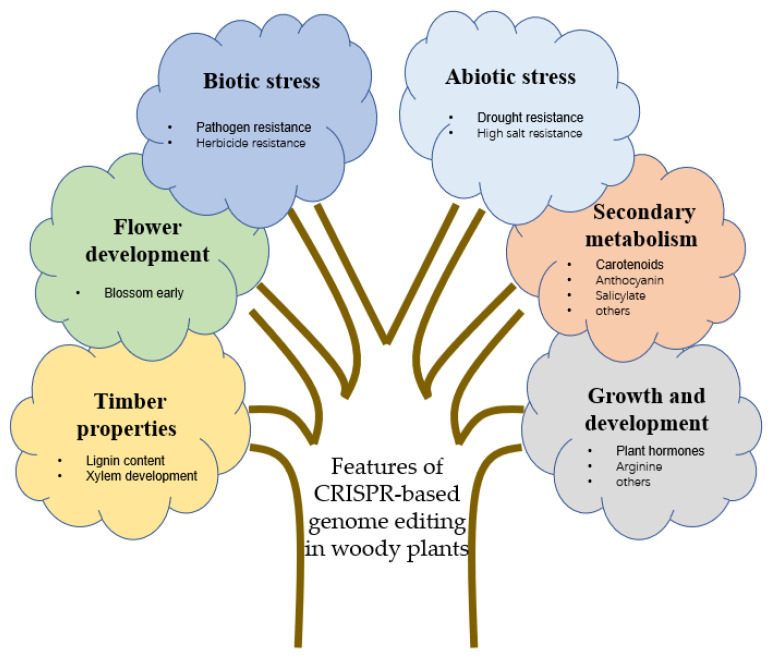
Features of CRISPR-based genome editing in woody plants (Provided by Microsoft PowerPoint 2019).

## Data Availability

Not Applicable.

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
