# Peer review of "CRISPR-Based Genome Editing and Its Applications in Woody Plants"

_ijms, 2022, doi:10.3390/ijms231710175_

Round 1

Reviewer 1 Report

The review article titled ‘CRISPR-based genome editing and its applications in woody plants’ by Min et al., is an interesting and extensive work. It presents the details of novel CRISPR/Cas based genome editing techniques and their applications employed in woody plants to improve the essential traits like growth, fruiting, flowering, biotic and abiotic stress, etc. Authors presented the novel CRISPR-Cas based editors, which has also provides a base for future studies of genetic advancements in woody plants. The review article fit well with the scope of the journal the manuscript merits the publication in International Journal of Molecular Sciences. However, the following corrections are required.

  1. I feel the presentation and structure of the article can be improved more.
  2. English writing can be improved.
  3. The meaning of the sentence ‘Regarding creating… in Line-117-118 is not clear. Please rewrite.
  4. Applications of many Cas12 variants in  plants is not mentioned.
  5. Aome of the images in Figure. 1 are not clearly visible. Please change.
  6. ‘Realizing’ should be replaced by ‘releasing’ in Line-294.
  7. ‘in vitro’ needs to be italicized in all places.
  8. Quote the reference for the work in Line-313.
  9. Title in the Section 4.4 can be ‘improvement stress resistence’.
  10. Meaning of the sentence CRISPR/Cas-mediated focused.. in Line-594 is not clear.
  11. Plant species and gene names in all places need to be italicized.

Author Response

Dear Reviewer and Editor,

Thanks very much for your time and efforts in handling and reviewing our manuscript. Your valuable comments and suggestions helped us to improve significantly the quality of the manuscript. We revised the manuscript according to your nice comments and suggestions and tried to improve the quality of the manuscript. Moreover, point-to-point responses were made and listed below.

Point 1:I feel the presentation and structure of the article can be improved more.

Response: Thank you for your valuable comments and suggestions. It helps improve the quality of our manuscript. The presentation and structure of the manuscript were improved as per your advice, headings and subheading were restructured, and other information was added to edify the quality of the manuscript. Please see the manuscript. 

Point 2:English writing can be improved.

Response: We appreciate your valuable suggestion, and we apologize for the state of our initial submission. However, English writing was professionally revised, and grammatical mistakes and expressions were proficiently improved accordingly. Please see the manuscript.

Point 3:The meaning of the sentence ‘Regarding creating… in Line-117-118 is not clear. Please rewrite.

Response Thank you very much for this observation and suggestion. The sentence in Line 117-118 was rephrased and paraphrased to convey the intended meaning and easy understanding by readers. Please see Line 117-118.  

Point 4:Applications of many Cas12 variants in plants is not mentioned.

Response: We appreciate your valuable observation in response to this point. We humbly agree with this observation, and we will maintain this state due to the fact that, according to our research, there is little to no mention of the application of Cas12 in plants. Hence, we have decided to skip its discussion. I hope you find our decision satisfactory.

Point 5:Some of the images in Figure. 1 are not clearly visible. Please change.

Response: Thank you very much for your observation of the quality of the first submitted figures. We have revised Figure 1 and changed it to a clearer high-quality figure. Please see the Figure in the manuscript.

Point 6:‘Realizing’ should be replaced by ‘releasing’ in Line-294.

Response: We apologize for this negligence in typos and appreciate your comment. However, the word was replaced as advised. Please see Line 294.

Point 7:‘in vitro’ needs to be italicized in all places.

Response: Thank you for this suggestion. We have implemented this suggestion, and the word in vitro has been changed to italics. Please see the manuscript.

Point 8:Quote the reference for the work in Line-313.

Response: Thank you for this valuable comment. We apologize for this error. However, this information is related to the previously cited work of Maher et al. [51], and the sentence has been corrected.

Point 9:Title in the Section 4.4 can be ‘improvement stress resistence’.

Response: Thank you for this valuable suggestion. However, after subheading restructuring as per your previous advice, we have decided to combine the initial subsection 4.4 with the initial subsection 4.3. to formulate a new subsection entitled Improvement of stress resistance. Please see Section 4.3 

Point 10:Meaning of the sentence CRISPR/Cas-mediated focused.. in Line-594 is not clear.

Response: Thank you for this comment. The sentence initially in Line 594 has been rephrased to convey the intended meaning and easy understanding for readers. Please see the manuscript, Line 762.

Point 11:Plant species and gene names in all places need to be italicized.

Response: We apologize for negligence in the first submission and thank you for this observation. Plant species and gene names have been changed to italics format. Please see the manuscript.

Reviewer 2 Report

  ​The review is meant to concentrate on editing in woody species. Actually the authors spend almost half of the text in reporting about recent innovation on new editors. This part is significantly too detailed and it does not add much to woody plants biotechnology (the examples reported are not even from the plant kingdom, and the new editors described are not used - besides a couple of cases - in woody plants). This long part should be drastically reduced or eliminated at once. The base editing part is seriously affected by conceptual mistakes (which are reflected in the figure) and should be rewritten. It is not immediately clear why the authors included the prime editing part, as this is not a technology with relevance for woody plants. If the authors want to keep this part, they should discuss why this technology is not yet used in woody plants. The paragraphs of part 4 (applications of gene editing) are confusing in that it is not clear the rationale which has been used in classifying the examples (plant development seems to be a category, but then several examples in this domain are reported under different categories, to make an example). The authors should recategorize the example and rewrite section 4. Another example is the listing of carotenoids under the category "growth and development", or the separate listing of stem branching (into the acclimatization category instead of growth), which is confusing. Also, the authors should make an effort to briefly introduce the scientific/business question which is underlying the examples they provide, in order to make the reader understand the context in which the editing has been applied. Many of the genes used as examples are merely reported as names, and the reader is left with little information as to why they are reported. The points addressed in the conclusion section are correct, although they are not extensively addressed. In general, the authors should make an effort to discuss why the technologies they describe have not been already used in woody plants (they mention it, but a thorough discussion is missing).  The text is affected by several mistakes, misspelling, grammar error, logical slips, ​which are reported here in a non-exhaustive manner. Because of the nature and quantity of mistakes, we strongly suggest that the authors seek the consultancy of a professional text editor. line (L) 16: sentence not clear, rephrase (R) L17: summarizes L19: correct typing mistakes (T) L 26: arbors is incorrect ​L27: essential and fundamentals are redundant ​L32: the authors are here restricting to quality; this is not what they describe in the rest of the text L37 et segg: not all are natural characteristics L44: R L47: regarding is incorrect L48-51: R L52: citation missing on the three systems L63: to state that TALES is similar to ZFN in composition is not clear and unexplicative L67-68: R L70: the Cas protein is not based on sgRNA, what do the Authors mean? L76: authors should homogenize the use of capital letters (i.e. sgRNA and not SgRNA); Genome editing does not need the capital L83: R L86: T L88: R There are no comments until section 3, as it has been indicated that this part is not targeted to the scope of the review and should be entirely rethought. L284-286: summarize and rewrite; make sure that the examples are from woody crops (in fact, most of the examples reported in this section are from other crops).  L308: WUS2 L315: DNA is not transmitted L 345 et segg: R L347: cellulose statement is unclear L370: what does condensed mean in this context? same with dictated L376 L377: R L394: Flowering LOCUS L408 et segg: R L 411: R L417: R L423: the logical link between the two parts of the sentence is missing L452: R L465 et segg: R L 483: R L494: R (also note that weeds are not abiotic stresses) L531 et segg: it is not clear how the mentioned technologies have been used, on which targets L535: this is redundant L538: detected is not properly used L562: R L566: exhaploid Figure 4 is not adding much knowledge, and it would benefit from the introduction of the list of some of the major genes being targeted The Conclusion (more appropriately Conclusions) section has to be rewritten, as it contains many mistakes and the sentences are not clearly written. In the Table, the common name of the plants should go along with the latin name, to avoid misunderstandings. The measure of the editing efficiency should be specified. The sorting criterion used to create the categories should be the same adopted in the revised manuscript.  Terms and words should be double checked (eg. canker instead of ulcer)   Throughout the text: genus and species should be italicized when in latin; the common name of species (i.e. poplar) does not need the capital; the use of the adverbs (i.e. "however") should be checked and the authors should make sure that the adverbs are correctly used in the context. All the gene names (and acronyms) and loci should be carefully checked and used when necessary.

Author Response

Dear Reviewer and Editor,

Thank you very much for your time and efforts in handling and reviewing our manuscript. Your valuable comments and suggestions helped us to greatly improve the quality of manuscript. We revised the manuscript according to your nice comments and suggestions and tried to improve the quality of manuscript. And point-to-point responses were made and listed below.

Point 1:The review is meant to concentrate on editing in woody species. Actually, the authors spend almost half of the text in reporting about recent innovation on new editors. This part is significantly too detailed and it does not add much to woody plants biotechnology (the examples reported are not even from the plant kingdom, and the new editors described are not used - besides a couple of cases - in woody plants). This long part should be drastically reduced or eliminated at once. The base editing part is seriously affected by conceptual mistakes (which are reflected in the figure) and should be rewritten. It is not immediately clear why the authors included the prime editing part, as this is not a technology with relevance for woody plants. If the authors want to keep this part, they should discuss why this technology is not yet used in woody plants.

Response: We are really grateful for this comment and observation. It’s really valuable. Second, we apologize for the poor articulation of the first section and the insignificant information added in the first section. However, we have implemented all the suggestions added to this comment. The recent innovation and new editors were paraphrased, and a greater part of it was omitted, leaving only a brief description of CRISPR/Cas12, CRISPR/Cas13, CRISPR/Cas14, base editing to lay the foundation of manuscript discussion focus. In addition, Figure 1 was revised to suit the review discussion. Please see the manuscript, Section 2, and Figure 1.

Point 2:The paragraphs of part 4 (applications of gene editing) are confusing in that it is not clear the rationale which has been used in classifying the examples (plant development seems to be a category, but then several examples in this domain are reported under different categories, to make an example). The authors should recategorize the example and rewrite section 4. Another example is the listing of carotenoids under the category "growth and development", or the separate listing of stem branching (into the acclimatization category instead of growth), which is confusing. Also, the authors should make an effort to briefly introduce the scientific/business question which is underlying the examples they provide, in order to make the reader understand the context in which the editing has been appl Many of the genes used as examples are merely reported as names, and the reader is left with little information as to why they are reported.

Response: We value this comment very much. It improves information delivery in our manuscript. We have implemented all the suggestions and comments underlined on this point. First, we have restructured Section 4 subsections and reassigned some contents to fitting subsections to allow readers for easy comprehension. Please see section 4 for more details.

Second, we introduced scientific/business questions in most of the applications we have discussed under Section 4. For instance, carotene in secondary metabolism and stem branches in growth and development, and each reported gene and the scientific or commercial problems were briefly introduced and further explained. We hope you will find these revisions to your satisfaction.

Last, reported genes from previous researches were briefly introduced, and their plant species were summarized, adding more information for readers’ understanding. Please see the manuscript.   

Point 3:The points addressed in the conclusion section are correct, although they are not extensively addressed. In general, the authors should make an effort to discuss why the technologies they describe have not been already used in woody plants (they mention it, but a thorough discussion is missing).  

Response: We thank you for this compliment and an additional suggestion. However, we have expanded the Conclusion section and tried to give our reasons some technologies described have not already been used in woody plants. We hope this conclusion expansion will be to your satisfaction. Please see the Conclusion in the manuscript.

Point 4:The text is affected by several mistakes, misspelling, grammar error, logical slips, ​which are reported here in a non-exhaustive manner. Because of the nature and quantity of mistakes, we strongly suggest that the authors seek the consultancy of a professional text editor. line (L) 16: sentence not clear, rephrase (R).

Response: We apologize for the poor English expression state of our first submission. However, we have consulted for professional English revisions. Grammar mistakes and several typos within the manuscript have been corrected, and Line 16 was rephrased to convey the intended meaning. Please see the manuscript.

Point 5: L17: summarizes

Response: Thank you for this valuable comment. Line 17 was summarized as advised. Please see Line 18. 

Point 6: L19: correct typing mistakes (T)

Response: We apologize for these mistakes. Typos in Line 19 were rectified as advised.  

Point 7: L 26: arbors is incorrect

Response: We are grateful for this comment and observation, arbors in Line 26 was replaced with macro-phanerophytes. Please see the manuscript.

​Point 8: L27: essential and fundamentals are redundant

Response: Thank you for this observation. The two words were omitted, and the entire sentence was rephrased. Please see Line 27 in the manuscript. The two words have been deleted.

​Point 9: L32: the authors are here restricting to quality; this is not what they describe in the rest of the text

Response: Thank you for this comment. Line 32 has been rephrased not to limit this discussion of quality only. Please see Line 31 in the manuscript.

Point 10: L37 et segg: not all are natural characteristics

Response: We apologize for this mistake. Line 37 was rephrased to include natural characteristics only, and biological characteristics were kept.

Point 11: L44: R

Response: Thank you for this comment. Line 44 was rephrased as advised.  

Point 12: L47: regarding is incorrect

Response: Thank you for this observation. The word regarding in Line 47 was replaced, and the whole sentence was rephrased. Please see Line 48. 

Point 13: L48-51: R

Response: Thank you for this comment. Line 48-51 was rephrased as follows; The recently developed gene editing technology can realize the precise editing of specific gene loci and shorten the breeding cycle. Therefore, the emergence of gene editing technology provides new ideas and ways for the genetic improvement of woody plants and the research of plant gene functions.

Point 14: L52: citation missing on the three systems

Response: We apologize for this omission; the citation was added to the three systems. 

Point 15: L63: to state that TALES is similar to ZFN in composition is not clear and unexplicative

Response: We apologize for this error. Nonetheless, the entire sentence has been removed from the manuscript because it was not significant in this discussion.

Point 16: L67-68: R

Response: Thank you for this advice. Line 67-68 was rephrased as follows; The Clustered Regularly Interspaced Short Palindromic Repeats/CRISPR-associated 9 (CRISPR/Cas9) system, which was first discovered in bacteria and archaea, is the third-generation gene editing technology, and it shows strong gene editing ability.

Point 17: L70: the Cas protein is not based on sgRNA, what do the Authors mean?

Response: We apologize for failing to deliver the intended meaning in this first submission. However, were to have rephrased this whole sentence to, ‘The CRISPR/Cas comprises CRISPR sequences and a Cas protein, in the process of genome editing with CRISPR/Cas9, single-guide RNA (sgRNA) formed by pairing CRISPR RNA (crRNA) and trans-activating crRNA (tracrRNA), which plays a role of targeted cutting.’

Point 18: L76: authors should homogenize the use of capital letters (i.e. sgRNA and not SgRNA); Genome editing does not need the capital

Response: We apologize for this mistake. Capital letter use has been homogenized, and proper wording has been implemented. Please see Line 86.

Point 19: L83: R

Response: Thank you for this comment. Line 83 has been rephrased as follows; Therefore, CRISPR/Cas technique is a milestone in genome editing technology and has various functions: increasing the proficiency of woody plant breeding, breeding cycles, diversifying woody plant traits, and regulating insufficiencies of conventional woody plant breeding.

Point 20: L86: T

Response: We value this comment. Line 86 has been corrected as advised. Please see the manuscript.

Point 21: L88: R

Response: Thank you for this comment. Line 88 has been rephrased as follows; In this review, the existing research on gene editing technology of woody plants was reported, and the limitations of its application in woody plant breeding and its future development trend were discussed.

Point 22: There are no comments until section 3, as it has been indicated that this part is not targeted to the scope of the review and should be entirely rethought. L284-286: summarize and rewrite; make sure that the examples are from woody crops (in fact, most of the examples reported in this section are from other crops). 

Response: We apologize for the state of the manuscript up to Section 3. The contents were rethought, and newly structured sections were added. Please see the manuscript until section 3. Line 284-286 was revised and unrelated crops related to woody plants were removed. However, a few plants not relating to woody plants were kept in order to summarize the common delivery methods existing plants that we hope may also be utilized in woody plants in the future. We also hope that our decision will be to your satisfaction.

Point 23: L308: WUS2

Response: Thank you for this comment. WUS2, in Line 308, has been corrected. Please see the manuscript.

Point 24: L315: DNA is not transmitted

Response: We apologize for this error. Correct information regarding DNA has been added to the manuscript.  

Point 25: L 345 et segg: R

Response: Thank you for this suggestion. Line 345 has been rephrased as follows; As a renewable biological material, wood is a necessity for human life. The formation and thickening of secondary the secondary cell wall are critical steps in wood formation and growth.

Point 26: L347: cellulose statement is unclear

Response: Thank you for this observation. This statement has been revised and corrected, and meaning has been added for reader’s sake. Please see the manuscript. 

Point 27: L370: what does condensed mean in this context? same with dictated

Response: We are grateful for this question. In Line 370, the words ‘reduced’ and ‘dictated’ have been used as synonyms of compacted and formed, respectively.

Point 28: L376 L377: R

Response: Thank you for this comment. Line 366-77 have been rephrased as follows; secondary cell walls were not formed in xylem wood fibers, xylem ray parenchyma cells, and phloem fibers in the four mutants of vns09/vns10/vns11/vns12, but only some wood fibers near vessel element showed secondary cell wall deposition.

Point 29: L394: Flowering LOCUS

Response: We apologize for this mistake. The Flowering LOCUS has been corrected. Please see the manuscript.

Point 30: L408 et segg: R

Response: Thank you for this comment. Line 408 has been deleted from the revised version of the manuscript. Due to the fact it was not significant to the intended meaning. 

Point 31: L 411: R

Response: Thank you for this comment. Line 411 has been deleted from the revised version of the manuscript. Due to the fact this example is not a woody plant.

Point 32: L417: R

Response: Thank you for this comment. Line 417 was rephrased as follows; Chang et al. [98] used CRISPR/Cas9 to knock out the two UDP-glycosyltransferase genes PgUGT84A23 and PgUGT84A24 in pomegranate, thus reducing the content of phenolic substance angoranin in pomegranate.

Point 33: L423: the logical link between the two parts of the sentence is missing

Response: We apologize for the poor delivery. Nonetheless, the two parts of the sentence have been revised as follows; Ren et al. [99] knocked out the tartaric acid (TA) synthesis key enzyme IdnDH in grape by using CRISPR/Cas9, which remarkably reduced the TA content in the mutant and confirmed the feasibility of applying the technology to grape genome transformation.’

Point 34: L452: R

Response: Thank you for this comment. Line 452 has been revised as follows; The mutant strain is more sensitive to Botrytis cinerea, which proves that VvPR4b has the disease resistance to Botrytis cinerea in grapes.

Point 35: L465 et segg: R

Response: Thank you for this comment. Line 465 has been revised as follows, and the probability of resulting biallelic mutants is lower compared to CRISPR/Cas9 editing.

Point 36: L 483: R

Response: Thank you for this comment. Line 483 has been revised as follows; Populus can recruit the abscisic acid (ABA) signaling pathway to adapt to a saline environment. The result found that PalWRKY77 negatively regulates the ABA-responsive gene and relieves ABA-mediated growth inhibition, which causes Poplar to be more sensitive to salt stress.

Point 37: L494: R (also note that weeds are not abiotic stresses)

Response: We are grateful for this observation. The sentence has been revised as follows:  Malabarba et al. [82] used base editing to create an amino-acid substitution on ALS genes in apple and pear, which resulted in the non-binding of the herbicide chlorsulfuron, so the mutant obtained resistance to the herbicide chlorsulfuron.

In addition, weeds have been classified as biological stressors of woody plants, and I hope our classification will be to your satisfaction.

Point 38: L531 et segg: it is not clear how the mentioned technologies have been used, on which targets

Response: Thank you for this comment. Line 531 has been revised as follows; The CRISPR/Cas9 technology was applied to knock out the PDS gene to produce albino plants used in sweet orange [84], poplar [85], apple [86], grape [87], cassava [88], Citrus [89], coffee [90], cotton [91], green bamboo [92], walnut [93] and other species.

Point 39: L535: this is redundant

Response: Thank you for this comment. Line 535 has been deleted from the revised version of the manuscript. Due to the fact that its information was insignificant to the discussion of woody plants.

Point 40: L538: detected is not properly used

Response: Thank you for this observation. The word' detected' was replaced as follows; ‘Cai et al. [101] knocked out cytokine synthesis related gene JcCYP735A of Jatropha curcas by CRISPR/Cas9 and decreased cytokinin content in leaves. The plants indicated apparent growth retardation, and the height was only a quarter of the wild plants.’

Point 41: L562: R

Response: Thank you for this comment. Line 562 has been deleted from the revised version of the manuscript. Because, its information was insignificant to the discussion of woody plants.

Point 42: L566: exhaploid Figure 4 is not adding much knowledge, and it would benefit from the introduction of the list of some of the major genes being targeted.

Response: We apologize for the poor illustration of the figure information. However, Figure 4 has been modified to include relevant information. Please see Figure 4.

Point 43: The Conclusion (more appropriately Conclusions) section has to be rewritten, as it contains many mistakes and the sentences are not clearly written.

Response: We apologize that our conclusion section was not to your satisfaction. However, the English was professionally revised, and mistakes were corrected. We also added some information to enrich the quality of this section. Please see the conclusion section.

Point 44: In the Table, the common name of the plants should go along with the latin name to avoid misunderstandings. The measure of the editing efficiency should be specified. The sorting criterion used to create the categories should be the same adopted in the revised manuscript. Terms and words should be double-checked (eg. canker instead of ulcer).

Response: We are grateful for this suggestion and advice. In Table 1, plant names were aligned with their Latin names, and the measure of the editing efficiency was specified. In addition, the sorting criteria was made consistent, and terms were double-checked. Please see the manuscript.

Point 45: Throughout the text: genus and species should be italicized when in latin; the common name of species (i.e. poplar) does not need the capital; the use of the adverbs (i.e. "however") should be checked and the authors should make sure that the adverbs are correctly used in the context. All the gene names (and acronyms) and loci should be carefully checked and used when necessary.

Response: We apologize for these mistakes. Genus and species and species names were italicized, and adverb use was corrected professionally. Furthermore, all the gene names and loci were carefully checked and used when relevant. Please see the manuscript.